# Lung Function and Respiratory Muscle Adaptations of Endurance- and Strength-Trained Males

**DOI:** 10.3390/sports8120160

**Published:** 2020-12-10

**Authors:** Daniel A. Hackett

**Affiliations:** Exercise, Health and Performance Faculty Research Group, School of Health Sciences, Faculty of Medicine and Health, The University of Sydney, Lidcombe, NSW 2141, Australia; daniel.hackett@sydney.edu.au; Tel.: +61-2-9351-9294; Fax: +61-2-9351-9204

**Keywords:** respiratory mouth pressures, respiratory muscles, resistance training, muscle strength, exercise performance

## Abstract

Diverse exercise-induced adaptations following aerobic endurance compared to strength-training programs is well documented, however, there is paucity of research specifically focused on adaptations in the respiratory system. The aim of the study was to examine whether differences in lung function and respiratory muscle strength exist between trainers predominately engaged in endurance compared to strength-related exercise. A secondary aim was to investigate if lung function and respiratory muscle strength were associated with one-repetition maximum (1RM) in the strength trainers, and with VO_2_ max and fat-free mass in each respective group. Forty-six males participated in this study, consisting of 24 strength-trained (26.2 ± 6.4 years) and 22 endurance-trained (29.9 ± 7.6 years) participants. Testing involved measures of lung function, respiratory muscle strength, VO_2_ max, 1RM, and body composition. The endurance-trained compared to strength-trained participants had greater maximal voluntary ventilation (MVV) (11.3%, *p* = 0.02). The strength-trained compared to endurance-trained participants generated greater maximal inspiratory pressure (MIP) (14.3%, *p* = 0.02) and maximal expiratory pressure (MEP) (12.4%, *p* = 0.02). Moderate–strong relationships were found between strength-trained respiratory muscle strength (MIP and MEP) and squat and deadlift 1RM (r = 0.48–0.55, *p* ≤ 0.017). For the strength-trained participants, a strong relationship was found between MVV and VO_2_ max (mL·kg^−1^·min^−1^) (r = 0.63, *p* = 0.003) and a moderate relationship between MIP and fat-free mass (r = 0.42, *p* = 0.04). It appears that endurance compared to strength trainers have greater muscle endurance, while the latter group exhibits greater respiratory muscle strength. Differences in respiratory muscle strength in resistance trainers may be influenced by lower body strength.

## 1. Introduction

Aerobic training is well known to induce structural and functional adaptations of the cardiovascular and musculoskeletal systems [1]. During exercise, increased stress is placed upon the respiratory system to meet the metabolic demands of the activity. The respiratory system consists of airways, lungs, blood vessels, and muscles. This network of organs and tissues is responsible for gas exchange between inspired air and the circulatory system where oxygen is delivered to the blood, as well as the elimination of carbon dioxide from the blood to the lungs [1]. Improvements in lung function have been shown in healthy inactive women following aerobic exercise combined with resistance training [2]. Improvements in respiratory muscle strength have also been observed following a 4-week high-intensity interval training program [3] and a 16-week resistance training program involving sit-ups and bicep curls [4]. The positive effects of exercise on the respiratory system are not limited to healthy populations, with positive effects found in adults with asthma [5], chronic stroke patients [6], and obese adolescents [7]. Therefore, it is not surprising that higher levels of aerobic fitness are associated with enhanced lung function [8] and a slower decline in lung function during aging with a more physically active lifestyle [9]. In athletic populations, the intense exercise training regimens they typically undergo appears to contribute to enhanced lung function compared to healthy sedentary adults [10] or age-matched reference values [11]. However, lung function between athletes has been shown to vary depending on the sport, with superior performance in endurance-trained athletes compared to athletes involved in strength/power training [10].

Athletes that are considered to be endurance trained predominantly engage in aerobic exercise which is defined as an activity that uses large muscle groups, can be maintained continuously for extended periods, and is rhythmic in nature (e.g., cycling, running, and swimming) [12]. This type of exercise heavily relies on aerobic metabolism, with maximal oxygen uptake (VO_2_ max) being a criterion measure of aerobic fitness [13]. Aerobic exercise is associated with greater ventilatory responses compared to resistance exercise, which is the predominant training type performed by strength/power athletes [14]. The great demands placed on the respiratory system during intense aerobic exercise have been documented in runners, with lung function being acutely impaired following marathon and ultra-marathon events [15]. In comparison, there does not seem to be any detrimental acute change in lung function following resistance training [16] and this lack of training stimulus on the lungs likely explains the similar lung function observed between strength/power athletes and sedentary adults [10]. However, unique respiratory system adaptations appear to take place within strength-trained athletes, as reported by Brown et al. [17], with greater diaphragm mass and respiratory muscle strength in world-class male powerlifters compared to untrained healthy adults. Supposedly, compound exercises such as squats and deadlifts may provide a stimulatory effect on the respiratory muscles due to their activation to assist with spine stability [18].

To date, there is a paucity of research that has investigated both the lung function and respiratory muscle strength in a given cohort comprising endurance-trained and strength-trained participants. As such, the lung function and respiratory muscle strength characteristics of these two different athletic populations have not been confirmed. Additionally, most studies have included a broad variety of athletes characterised as endurance trained (distance runners, footballers) or strength trained (short-distance runners, wrestlers, weightlifters, martial art fighters) [10,11], which is likely to influence the results due to the engagement in different types of training. This is especially the case for the strength-trained groups since the resistance-training stimulus (e.g., loads, type of exercise, training volume) appears to be important towards inducing respiratory muscle adaptations [4,17]. Therefore, the purpose of this study was to examine whether differences in lung function and respiratory muscle strength exist between trainers that predominately engage in endurance or strength-related exercise. A secondary aim was to investigate if lung function and respiratory muscle strength were associated with one-repetition maximum (1RM) in the strength-trained group and with VO_2_ max and fat-free mass in each group. It was hypothesised that superior performance in the endurance-trained compared to strength-trained group would be observed for indices of lung function, which is supported by evidence of enhanced lung function in athletes with endurance training experience [10,11]. Greater respiratory muscle strength was expected in the strength-trained compared to endurance-trained group based on the findings from Brown et al. [17]. Further, it was expected that lung function would be associated with VO_2_ max and that respiratory muscle strength would be associated with 1RM [19] and fat-free mass [20].

## 2. Materials and Methods

A cross-sectional, descriptive, and correlational study design was employed to examine the lung function and respiratory muscle strength of strength-trained and endurance-trained males. Forty-six males participated in this study, consisting of 24 strength-trained (26.2 ± 6.4 years; 1.8 ± 0.1 m) and 22 endurance-trained (29.9 ± 7.6 years; 1.8 ± 0.1 m) participants. Participants in the strength-trained group reported 7.4 ± 5.3 years of resistance training experience and would complete this type of training on 4.0 ± 1.1 days per week. There were 18 participants that were considered recreational resistance trainers, five participants that had a background in powerlifting/Olympic weightlifting, and one participant with a history of competing in bodybuilding contests. In comparison, endurance-trained participants reported 8.3 ± 6.2 years of aerobic training experience and completing this training on 5.8 ± 3.4 days per week (>500 min per week). The predominant training of the endurance group consisted of cycling (*n* = 8), triathlon (*n* = 7), running (*n* = 6), and cycling/running (*n* = 1). Participants provided verbal and written consent prior to study commencement. This study was approved by the University of Sydney Human Research Ethics Committee, project number 2014/996.

The general eligibility criteria for this study included being male, aged 18–45 years, and apparently healthy, which was defined as the absence of asthma, current illness, medications that influence lung function, and any chronic disease or condition. While pulmonary function and aerobic capacity declines between the ages of 25–80 years, an age of 45 years was considered the upper limit where the effects of age would have minimal influence on the respiratory and exercise performance of an active cohort [21]. The specific eligibility criteria for the strength-trained group included: ≥1 year resistance training experience with ≥2 sessions per week currently being performed, and having the ability to perform the bench press, squat, and deadlift. A specific strength level was also required to be met by participants in the strength group, which included a relative muscle strength (1RM kg/kg body mass (BM)) of ≥1.2 for the bench press, ≥1.5 for the squat, and ≥1.7 for the deadlift. The specific eligibility requirements for the endurance-trained group included consistent participation in aerobic exercise training sessions (averaging ≥3 sessions per week) featuring activities such as cycling, running, and swimming. To ensure that the endurance group was adequately trained, a VO_2_ max of >50.0 mL·kg^−1^·min^−1^ was required since physically active but not highly trained males can be shown to achieve VO_2_ max values of 50.1 ± 3.1 mL·kg^−1^·min^−1^ [22].

Participants were tested at the Exercise Physiology Laboratories of The University of Sydney. For the strength-trained group, this involved two to three visits with a cycling VO_2_ max test performed during the initial session, the lung function and respiratory muscle tests in the second session, and the one-repetition maximum (1RM) as the last test to be performed. Body composition was either assessed in a separate visit or at the beginning of other visits. The endurance-trained group completed all testing in one visit in the order of body composition, lung function and respiratory muscle strength testing, and finally the cycling VO_2_ max test. It was decided that the endurance-trained group would not be involved in 1RM testing due to the general lack of resistance training experience. There were 12/22 participants who did not engage in any resistance training, 1/22 participants that performed bodyweight resistance exercise once per week, and 9/22 participants who performed 1–2 days of resistance training. Additionally, it was likely that having novice resistance trainers perform the squat and deadlift 1RM may expose the endurance-trained participants to unnecessary risks and could confound the results due to the highly technical nature of these lifts. Participants were instructed to avoid any strenuous physical activity 24–48 h before the testing session. If a participant reported fatigue or soreness from previous exercise, the testing session was rescheduled. If performing exercise testing in the same visit as the body composition assessment, it was advised to eat a light meal following a dual-energy X-ray absorptiometry (DEXA) scan so that exercise performance would not be negatively affected. Participants in the strength-trained group completed all testing in 9.0 ± 5.8 days.

### 2.1. Cycling VO_2_ Max Test

The VO_2_ max test was performed by participants exercising on an electronically braked cycle ergometer (Excalibur Sport V 2.0 bicycle, Lode BV, the Netherlands) with breath-by-breath recordings of VO_2_ throughout the test (Ultima Series CardiO2 and PFX, Medgraphics, Minneapolis, MN, USA). The protocol commenced at 100 W or 150 W, maintaining a cadence above 70 revolutions per minute with increases of 30 W every minute until exhaustion [23]. Heart rate measurements (Polar T31, Polar Electro Oy, Kemple, Finland) were recorded throughout the test. A 5–10 min warm-up was completed prior to commencing the VO_2_ max test. The VO_2_ values were averaged using a 10 s interpolation and the highest VO_2_ value prior to exhaustion was considered the VO_2_ max. The VO_2_ max was confirmed by the attainment of two or more of the following criteria: (1) an increase in work rate without an increase in VO_2_, (2) respiratory exchange ratio exceeding 1.1, and (3) heart rate within 10% of age-predicted maximum. The VO_2_ max was expressed in absolute (L·min^−1^) and relative (mL·kg^−1^·min^−1^) terms.

### 2.2. Body Composition

Participants had their body composition assessed by a whole-body DEXA scanner (Lunar Prodigy, GE Medical Systems, Madison, WI, USA). Conditions were standardised to ensure accurate results. This included being in a fasted state (10–12 h prior to scan), minimising fluid intake (no more than 200 mL of water prior to scan), bladder/bowel being voided, jewellery removed, and clothes removed down to the underwear with a hospital gown worn. Prior to every DEXA scan, the machine was calibrated (within 24 h). Body composition results were obtained by in-built analysis software (version 13.60.033; enCORE 2011, GE Healthcare, Madison, WI, USA).

### 2.3. Lung Function

The Medgraphics pulmonary function testing system (Breezesuite Ultima PFX, Milano, Italy) was used to assess lung function. Prior to all testing, participants were given information about the protocols for each test and were able to have practice attempts. The lung function measures were performed in a standing position. During the tests, participants wore a nose clip and were instructed to keep their lips securely around the mouth piece to prevent any air escaping. The first test performed was the forced vital capacity (FVC) test and required participants to empty their lungs of air and then fully inspire, followed by full expiration. The forced expiratory volume in 1 s (FEV_1_), 3 s (FEV_3_), and 6 s (FEV_6_), and ratio of FEV_1_ to FVC (FEV_1_/FVC) were obtained from the FVC. The slow vital capacity (SVC) test was performed after the FVC test and commenced with a minimum of four stable tidal breaths followed by a maximal inspiration and then a maximal expiration, performed in a slow manner. Both inspiratory capacity (IC) and expiratory reserve volume (ERV) were obtained from the SVC. A minimum of three trials were performed for the FVC and SVC tests, with the best two values needing to be within 5% [24]. The reliability between trials was assessed via the intraclass correlation coefficient (ICC) and coefficient of variation (CV). The reliability was considered good for the FVC (ICC = 0.99, 95% CI: 0.90–1.0; CV = 1.4%) and SVC (ICC = 0.99, 95% CI: 0.99–1.0; CV = 1.3%) tests.

The final lung function test was the maximal voluntary ventilation (MVV) test, with the participant required to breathe deeply and rapidly on the command over a period of 12 s. There was approximately a 30 s recovery between lung function testing trials, although generally there was a longer rest between attempts for MVV. At least three trials were performed of MVV, with the best two values needing to be within 10%. The best trial for all measures was used for data analysis. The reliability between trials was considered good for MVV (ICC = 0.99, 95% CI: 0.97–0.99; CV = 2.4%). Following adequate rest (i.e., >10 min), residual volume (RV) was assessed using the previously validated O_2_ rebreathing technique [25]. Participants performed two trials and these were averaged to determine RV. If the difference between trials 1 and 2 was greater than 300 mL, a third RV trial was required (with the two closest two trials averaged for the RV). The total lung capacity (TLC) was calculated through adding SVC (or FVC if greater) and RV.

### 2.4. Respiratory Muscle Strength

Maximal inspiratory pressure (MIP) and maximal expiratory pressure (MEP) were assessed using a handheld non-invasive mouth-pressure manometer that digitally displayed pressures on a small MicroRPM screen (Micro Medical/CareFusion, Kent, UK). Prior to each trial, participants wore a nose clip to prevent nasal air leak and were told to hold the device. The MIP was assessed first and involved emptying the lungs of air and then inhaling maximally for approximately 2–3 s against the resistance of the gauge. As for MEP, the participants fully filled their lungs with air and then exhaled maximally for approximately 2–3 s against the resistance of the gauge. There needed to be at least three trials within 10% of each other for MIP and MEP, with the best performances used for data analysis. Recovery between trials was approximately 30 s, although generally it was adjusted based on participants’ perceptions and variation in performances. The reliability between trials was considered good for MIP (ICC = 0.99, 95% CI: 0.98–0.99; CV = 3.8%) and MEP (ICC = 0.99, 95% CI: 0.98–0.99; CV = 3.6%).

### 2.5. Weightlifting Strength

The 1RM was assessed for the bench press, squat, and deadlift in participants from the strength-trained group. The order of exercise testing was either the bench press or squat first (participant could choose), with the deadlift assessed last. A thorough warm-up was completed prior to all 1RM tests. The warm-up involved performing sets using light loads and then progressing to heavier loads, although efforts were made to keep below approximately 80% of the maximum to reduce any possible fatigue effects. The 1RM test involved performing trials of a single repetition of increasing load with correct technique with a 3–5 min rest between attempts. The heaviest load successfully lifted was recorded as the 1RM. The technique used for the bench press involved lying flat on a bench, lowering the barbell close to chest (approximately 2.5 cm from chest) and then pressing upwards until arms were fully extended. For the squat, a successful attempt required the knees and hips to be in a horizontal plane (i.e., thighs parallel to floor) prior to commencing the ascent to an upright position. Finally, the deadlift technique involved the participant lifting the loaded barbell from the floor to a fully upright position with knees extended. Participants could not use weight belts for squat and deadlift but wrist straps and chalk were permitted to assist with holding the bar for the deadlift. The 1RM was expressed in absolute (kg) and relative terms (kg/kg BM).

### 2.6. Statistical Analyses

Data analyses were performed using SPSS version 24.0 for Windows (IBM Corp., Armonk, NY, USA). Normality of data was assessed through using the Kolmogorov–Smirnov test. The ICC was calculated using a two-way mixed model, absolute agreement, single rater/measurement [26]. Coefficient of variation (CV) was calculated by dividing the standard deviation (SD) by the mean of trials × 100. All data except bench press 1RM (kg/kg BM) were normally distributed. Differences between groups for all measures were assessed with an independent Student’s *t*-test. Estimates of effect size were calculated using standardised differences in means (mean difference divided by pooled standard deviation) and expressed as Hedges’ g, which corrects for parameter bias due to small sample sizes [27]. The magnitude of the effect sizes was assessed using the following criteria: trivial: ≤0.2, small: 0.21–0.59, moderate: 0.6–1.19, large: 1.2–1.99, very large: 2.0–3.9, and extremely large: ≥4.0 [28]. For the strength-trained group, the Pearson correlation coefficient was used to assess the relationships between respiratory muscle strength (MIP and MEP) and 1RM performance for bench press, squat, and deadlift (absolute and relative). The Spearman rank-order correlation coefficient, which is a non-parametric test, was used to assess the relationships with bench press 1RM (kg/kg BM). For each individual group (i.e., analyses conducted separately for endurance- and strength-trained groups), the Pearson correlation coefficient was used to assess the relationships between MVV and VO_2_ max (expressed in absolute and relative terms) and also between respiratory muscle strength and fat-free mass. Strengths of correlations were qualitatively assessed as: trivial (r < 0.1), small (r > 0.1 to 0.3), moderate (r > 0.3 to 0.4), strong (r > 0.5 to 0.7), very strong (r > 0.7 to 0.9), nearly perfect (r > 0.9), and perfect (r = 1.0) [28]. Data were presented as mean ± SD and the alpha level for significance was set at *p* <0.05.

## 3. Results

### 3.1. Characteristics of Participants

There was no significant difference between groups for age and height (*p* > 0.05). Table 1 provides the body composition and aerobic fitness characteristics of the groups. The strength-trained group had greater total body mass and fat-free mass (*p* = 0.001), while the endurance-trained group had lower fat mass and percentage of body fat (≤0.001). The endurance-trained compared strength-trained group had greater absolute VO_2_ max (L·min^−1^) (*p* < 0.002) and VO_2_ max (mL·kg^−1^·min^−1^) (*p* < 0.001). For the strength-trained group, the bench press 1RM was 115.2 ± 19.5 kg (1.4 kg/kg BM), the squat 1RM was 166.0 ± 31.6 kg (2.0 kg/kg BM), and the deadlift 1RM was 185.9 ± 34.3 kg (2.2 kg/kg BM).

### 3.2. Lung Function and Respiratory Muscle Strength

There was a trend towards statistical significance for the endurance-trained group having a greater TLC compared to the strength-trained group (*p* = 0.05) (Table 2). There were no differences between groups for FVC, FEV_1_, FEV_1_/FVC, FEV_3_, FEV_6_, SVC, IC, ERV, and RV. A significant difference between groups was found for MVV, with MVV for the endurance-trained group 11.3% greater compared to the strength-trained group (*p* = 0.04), resulting in a moderate effect size (ES = −0.68, 95% CI −0.03 to −1.33). There was a significant difference in MIP between groups, with MIP being 14.3% greater for the strength-trained compared to endurance-trained group (*p* = 0.02), resulting in a moderate effect size (ES = 0.75, 95% CI 1.34 to −0.15) (Figure 1A). The MEP was also significantly different between groups, with MEP being 12.4% greater for the strength-trained compared to endurance-trained group (*p* = 0.03), also resulting in a moderate effect size (ES = 0.65, 95% CI 1.24 to −0.06) (Figure 1B).

For the strength-trained group, significant moderate to strong positive relationships were found between both of the respiratory muscle strength measures (MIP and MEP) and 1RM (kg) for the squat and deadlift (r = 0.48–0.55, *p* ≤ 0.017) (Figure 2A–D). The MEP was found to be significantly related (strong relationship) to 1RM (kg/kg BM) for the squat (r = 0.55, *p* = 0.005; 95% CI: 0.19, 0.78) and deadlift (r = 0.55, *p* = 0.006; 95% CI: 0.19, 0.78). No significant relationships were found between MIP and 1RM (kg/kg BM) for the squat and deadlift. A significant strong positive relationship was found between MIP and bench press 1RM (kg) (r = 0.52, *p* = 0.01; 95% CI: 0.15, 0.76). No significant relationships were found between MIP and bench press 1RM (kg/kg BM) or between MEP and bench press 1RM (kg) and 1RM (kg/kg BM). When analysing the data of both groups, a significant moderate positive relationship was found between fat-free mass and MIP for the strength-trained group (r = 0.42, *p* = 0.04; 95% CI: 0.02 to 0.70), but not for the endurance-trained group (r = 0.35, *p* = 0.10; 95% CI: −0.08 to 0.67). There was no significant relationship found between fat-free mass and MEP for the strength-trained group (r = 0.28, *p* = 0.18; 95% CI: −0.14 to 0.61) or endurance-trained group (r = −0.13, *p* = 0.56; 95% CI: −0.31 to 0.52). For MVV, there was a significant strong relationship found with VO_2_ max (L·min^−1^) for the strength-trained group (r = 0.55, *p* = 0.009; 95% CI: 0.16 to 0.79) and endurance-trained group (r = 0.55, *p* = 0.02; 95% CI: 0.11 to 0.81). Relative VO_2_ max (mL·kg^−1^·min^−1^) was strongly related with MVV for the strength-trained group (r = 0.63, *p* = 0.002; 95% CI: 0.27 to 0.83) but no relationship was found for the endurance-trained group (r = 0.07, *p* = 0.80; 95% CI: −0.41 to 0.52). There were no significant relationships found between any other lung function indices and VO_2_ max.

## 4. Discussion

The purpose of this study was to examine whether differences in lung function and respiratory muscle strength exist within trainers that predominately engage in endurance or strength-related exercise. In agreement with the original hypothesis, the endurance-trained group displayed superior lung function, as indicated by an 11.3% greater maximal voluntary ventilation. Additionally, there was a trend approaching statistical significance for greater TLC in the endurance-trained compared to strength-trained group. In contrast to these lung function results, the strength-trained compared to endurance-trained group displayed greater respiratory muscle strength, which was also hypothesised. Specifically, the strength-trained group produced 14.3% greater maximal inspiratory pressure (MIP) and 12.4% greater maximal expiratory pressure (MEP). These respiratory muscle strength performances were moderately to strongly related with squat and deadlift 1RM in the strength-trained group. Fat-free mass also appeared to influence inspiratory muscle performance with a moderate relationship found for the strength-trained group. Strong relationships were also found between MVV and VO_2_ max (L·min^−1^) for the strength-trained and endurance-trained groups, however, this type of relationship remained only in the strength-trained group when expressed as relative VO_2_ max (mL·kg^−1^·min^−1^). Based on the present study findings, trainers that predominately engage in endurance or strength-related exercise display unique lung function and respiratory muscle strength characteristics. It appears that differences in respiratory muscle strength in resistance trainers may be influenced by lower body strength. However, it is unclear what factors may influence the greater respiratory muscle endurance observed in the participants that predominately engage in endurance training.

Athletes are known to possess enhanced lung function capacities compared to untrained healthy populations [10,29]. However, within athletic populations, there are distinct differences in the lung function characteristics, namely between athletes involved in endurance training (e.g., running, swimming, cycling) and strength/power athletes (e.g., short-distance runners, wrestlers, weightlifters, martial art fighters) [10,11]. Athletes that have endurance training experience have shown greater performance in FVC, FEV_1_, VC, and MVV [10,11]. Although the present study only showed that MVV was greater in the endurance-trained group, there were numerous lung function indices (i.e., FVC, FEV_6_, RV, TLC) that tended to favour the endurance-trained group, with *p* values between 0.05 to <0.10 and small effect sizes (−0.54 to −0.59). The possession of enhanced lung function in endurance-trained compared to strength-trained participants is likely the result of training adaptations to greater and prolonged ventilation to allow for meeting the gas exchange demands of the exercise. These high demands on the respiratory system during endurance training and events is reflected by occurrences of hypoxaemia is some athletes [30].

The finding of greater respiratory muscle strength in the strength-trained compared to endurance-trained group is novel in the sense that it was clearly shown that advanced resistance trainers have developed a unique respiratory system adaptation. Previously, Brown et al. [17] reported greater diaphragm mass and respiratory muscle strength in world-class male powerlifters compared to untrained healthy adults. Further, DePalo et al. [4] had subjects perform training sessions of sit-ups and biceps curls over 16 weeks and found significant increases in diaphragm thickness, resulting in greater inspiratory and expiratory muscle strength. Since the ventilatory demands of resistance training is generally lower compared to aerobic exercise [14], it is obvious that the training stimulus on the respiratory muscles differs for resistance exercise compared to aerobic exercise. During a resistance exercise, intra-abdominal pressure increases as a result of heavier loads used and greater fatigue levels (i.e., closer to momentary muscle failure) [31,32]. The increase in intra-abdominal pressure (i.e., increased pressurisation of the abdominal cavity) is achieved through contraction of the diaphragm, which moves this inspiratory muscle inferiorly acting on the relatively incompressible abdominal contents and is aided by coactivation of abdominal muscles [18,33]. Therefore, muscles involved in ventilation, including the diaphragm, thoracic cage muscles, and abdominal muscles, are recruited during resistance exercise to assist with the elevation of intra-abdominal pressure. It is therefore of no surprise that acute reductions in respiratory strength have been found following sit-ups [34].

Exercises performed by the strength-trained participants in the present study are used in powerlifting competitions and are commonly performed by experienced resistance trainers to maximise muscle adaptations [35]. Additionally, the squat and deadlift require high axial loading as well as increased lumbar spine stability compared to traditional resistance exercises (e.g., machine knee extension and flexion) [36]. As such, it seemed plausible that targeting participants well trained in the bench press, squat, and deadlift would provide a cohort that has strong “core” musculature, which, as mentioned previously, consists of the respiratory muscles. This would especially be true for weightlifting exercises with high axial loading, such as deadlifts and squats, where greater diaphragm activation has been reported [37]. The significant strong relationships between the respiratory muscle measures and 1RM for the squat and deadlift may suggest that resistance exercise, requiring greater lumbar stability, could provide a respiratory muscle strength-training stimulus. Previously, significant, although weaker, relationships compared to the present study were found between respiratory muscle strength and knee flexor and extensor muscle strength (r = 0.21–0.41) in a mixed athlete (e.g., judo, gymnastics) cohort [19]. However, the findings from the present study appear to be the first to document these relationships in more traditional weightlifting exercises that tend to differ in core stability requirements. It should be noted that the relationships between respiratory muscle strength and deadlift performance could have been impacted by some participants performing the squat 1RM immediately prior to the deadlift 1RM. In this case, the deadlift 1RM may have been negatively affected and influenced the associations with MIP and MEP. Additionally, some participants used wrist straps which, again, could have influenced the relationships with the respiratory muscle strength measures. This is based on evidence of increased deadlift 1RM through the wearing of wrist straps [38].

In the general population, lung function and respiratory muscle strength vary due to numerous factors, such as age, height, and body composition [20,39,40]. The age and height of participants in the endurance-trained and strength-trained groups in the present study were not significantly different. However, there were differences found for body weight, fat-free mass, and percentage of body fat. In particular, fat-free mass was significantly related to MIP in the strength-trained group (r = 0.42, 95% CI: 0.02 to 0.70), with a slightly weaker, although non-significant (*p* = 0.10), relationship found with the endurance-trained group (r = 0.35, 95% CI: −0.08 to 0.67). These findings are mostly in agreement with Ro et al. [20], where skeletal muscle index (relative to body mass) was related to MIP in healthy young adults. Fat-free mass is a major determinant of maximal strength and it will be generally greater in strength athletes compared to endurance athletes [41,42]. It is interesting that MEP was not correlated with fat-free mass in the present study. An explanation for this result could be due to the numerous muscles contributing to MEP (i.e., thoracic cage muscles and abdominal muscles) that may respond differently to training stimuli compared to the diaphragm, which is the main muscle used during MIP.

The MVV test is commonly used to assess the endurance of the inspiratory and expiration muscles [15]. However, performance in the MVV test can be influenced by many factors (e.g., respiratory system mechanics and respiratory muscle endurance) [43]. This best explains the improvement in MVV but no other lung function parameter following a resistance training intervention in healthy adults [2]. Since MVV was significantly correlated with relative VO_2_ max in the strength-trained (r = 0.63, 95% CI: 0.27 to 0.83) but not the endurance-trained group, it suggests that changes in respiratory muscle endurance may be of importance to aerobic exercise power for exercise trainers with lower VO_2_ max, i.e., approximately 35–40 mL·kg^−1^·min^−1^. Since exercise trainers with higher VO_2_ max have the ability to sustain high ventilation, it appears to be a product of exercise training type leading to specific respiratory system adaptations. However, the contribution of central and peripheral adaptations rather than respiratory system adaptations most likely influences the VO_2_ max (mL·kg^−1^·min^−1^) of endurance trainers [44]. Sufficient core stability is required during the squat and deadlift to enhance the performance of these exercises by efficiently transferring force through the kinetic chain, in addition to reducing the risk of injuries [45]. To assist with providing spinal stability during these exercises, weight belts are generally worn [46]. The weight belt elevates intra-abdominal pressure through aid to the core muscles. In the present study, to provide a clearer indication of the relationship between respiratory muscle strength and weightlifting 1RM, it was decided that participants could not use weight belts. It must also be emphasised that previous studies investigating the lung function and respiratory muscle performance of athletes have included a combined strength/power group [10,11]. The training performed by athletes involved in judo would be considerably different to powerlifters/weightlifters, thus probably resulting in different training stimuli on the respiratory system. A strength of the current study is the performance standards that needed to be met for inclusion in the endurance-trained group (i.e., VO_2_ max > 50.0 mL·kg^−1^·min^−1^) and the strength-trained group (relative 1RM thresholds).

## 5. Conclusions

In male trainers that predominately engage in endurance compared to strength-related exercise, there appears to be unique differences in respiratory system characteristics. The endurance-trained group exhibited superior respiratory endurance as well as a trend towards greater lung function performance for various indices compared to the strength-trained group. In contrast, the strength-trained group displayed greater performance in the respiratory muscle strength tests. Respiratory muscle endurance was strongly related to relative VO_2_ max for the strength-trained but not endurance-trained group. Differences in respiratory muscle strength in resistance trainers appear to be influenced by lower body strength. However, it is less clear what factors may influence the greater respiratory muscle endurance observed in endurance trainers.

## Figures and Tables

**Figure 1 sports-08-00160-f001:**
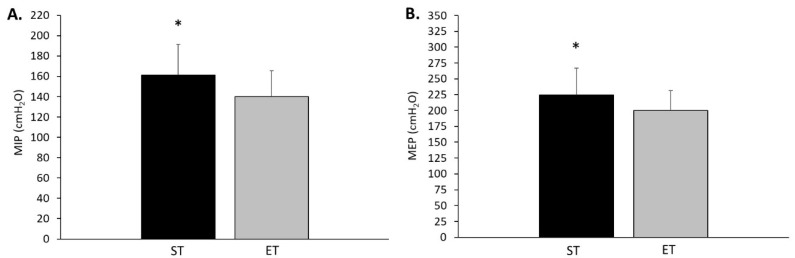
Respiratory muscle strength of endurance-trained (ET) and strength-trained (ST) participants. (**A**) Differences between groups in maximal inspiratory pressure (MIP); (**B**) differences between groups in maximal expiratory pressure (MEP). * Significant difference at *p* < 0.05.

**Figure 2 sports-08-00160-f002:**
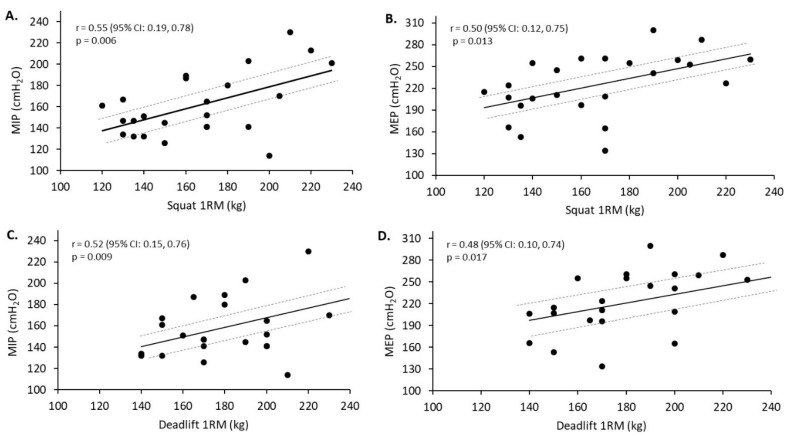
Relationships between respiratory muscle strength, squat one-repetition maximum (1RM), and deadlift 1RM. (**A**–**D**) Relationships between respiratory muscle strength (MIP—maximal inspiratory pressure; MEP—maximal expiratory pressure) and 1RM weightlifting performance (squat and deadlift) in the strength-trained group. Dashed grey lines represent the 95% confidence interval (CI).

**Table 1 sports-08-00160-t001:** Differences in body composition and aerobic fitness between the strength-trained and endurance-trained participants.

Variable	Strength Trained (*n* = 24)	Endurance Trained (*n* = 22)	*p* Value	ES (g)	95% CI
Body mass (kg)	85.09 ± 10.62	74.55 ± 8.20	0.001 *	1.09	1.71, 0.47
Fat-free mass (kg)	71.34 ± 8.52	65.62 ± 6.59	0.02 *	0.73	1.33, 0.14
Fat mass (kg)	13.93 ± 4.75	8.94 ± 3.34	<0.001 *	1.19	1.81, 0.56
Body fat (%)	16.85 ± 4.69	12.35 ± 3.91	0.001 *	1.02	1.64, 0.41
VO_2_ (L^−1^·min^−1^)	3.27 ± 0.56	4.30 ± 0.60	0.002 *	−1.75	−1.07, −2.43
VO_2_ (mL·kg^−1^·min^−1^)	38.67 ± 5.89	58.26 ± 5.17	<0.001 *	3.46	−2.55, −4.38

VO_2_—maximal oxygen uptake; CI—confidence interval; ES—effect size. Data reported as mean ± standard deviation (SD). * Significant difference at *p* < 0.05.

**Table 2 sports-08-00160-t002:** Comparison of lung function between strength-trained and endurance-trained participants.

Variable	Strength Trained (ST)	Endurance Trained (ET)	*p* Value	ES (g)	95% CI
FVC (L) ^a^	5.03 ± 0.71	5.41 ± 0.65	0.09	−0.55	0.09, −1.18
FEV_1_ (L) ^a^	4.22 ± 0.60	4.41 ± 0.67	0.36	−0.29	0.33, −0.92
FEV_1_/FVC (%) ^a^	83.92 ± 6.6	81.3 ± 6.6	0.22	0.39	1.02, −0.24
FEV_3_ (L) ^a^	5.0 ± 0.70	5.33 ± 0.67	0.13	−0.47	0.16, −1.10
FEV_6_ (L) ^a^	5.03 ± 0.71	5.41 ± 0.65	0.09	−0.55	0.09, −1.18
SVC (L) ^b^	5.29 ± 0.70	5.62 ± 0.70	0.13	−0.46	0.14, −1.06
IC (L) ^b^	3.80 ± 0.61	4.12 ± 0.76	0.14	−0.46	0.14, −1.06
ERV (L) ^b^	1.49 ± 0.51	1.50 ± 0.59	0.92	−0.02	0.57, −0.61
TLC (L) ^b^	6.52 ± 0.89	7.08 ± 0.99	0.05	−0.59	0.02, −1.19
MVV (L·min^−1^) ^c^	180.57 ± 28.23	200.94 ± 30.60	0.04 *	−0.68	−0.03, −1.33
RV (L) ^d^	1.22 ± 0.38	1.45 ± 0.46	0.07	−0.54	0.05, −1.13

^a^ ST (*n* = 21) and ET (*n* = 19); ^b^ ST (*n* = 23) and ET (*n* = 21); ^c^ ST (*n* = 21) and ET (*n* = 18); ^d^ ST (*n* = 24) and ET (*n* = 22). CI—confidence interval; ES—effect size; FVC—forced vital capacity; FEV_1_—forced expiratory volume in one second; FEV_3_—forced expiratory volume in three seconds; FEV_6_—forced expiratory volume in six seconds; SVC—slow vital capacity; IC—inspiratory capacity; ERV—expiratory reserve volume; TLC—total lung capacity; MVV—maximum voluntary ventilation; RV = residual volume. Data reported as mean ± standard deviation (SD). * Significant difference at *p* < 0.05.

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
