# Peer review of "Lung Function and Respiratory Muscle Adaptations of Endurance- and Strength-Trained Males"

_sports, 2020, doi:10.3390/sports8120160_

Round 1

Reviewer 1 Report

Please refer to the document attached. 

Author Response

Thank you for reviewing this manuscript and providing helpful feedback.

Please find below a point-by-point response to all the comments raised.

Abstract

  1. Please clarify the study aims. Particularly because you should remark that you are analysing the differences between two groups, and secondly, the associations between variables in two different groups.

Response: Thank you for this suggestion. The Abstract has now been amended.

  1. Revise your conclusion according to study findings.

Response: This has also now been revised.

Introduction

  1. Lines 30. Please clarify what type of training.

Response: ’aerobic’ training has been added.

  1. Lines 31-32. Repetitive at the start of both sentences.

Response: Now has been amended.

  1. Line 36-37. Please clarify. If this is following an interval training program or right after the session.

Response: Thank you for this recommendation. This has now been revised.

  1. Line 48 & 50. Aerobic exercise, is repetitive in both sentences.

Response: One of these sentences have been deleted so that the repetition of ‘aerobic exercise’ is now removed.

  1. Line 52. I think the correct word is ‘greater’.

Response: Thank you – this has been amended.

  1. Line 63. Please revise: ‘muscle strength in in a’.

Response: Corrected

  1. Line 64. It seems to me that you are not analysing changes or effects, but instead the characteristics of this two different athletic populations.... Please revise and reword.

Response: Has been revised as suggested.

  1. Line 66-67. It is also important to consider that the differences that the author is reflecting here, may be in part to the fact that weightlifters may not be engaged in endurance training protocols but in contrast to these endurance athletes needs to be engaged in strength training regimes as recommended in the literature. The same applies to other sports that are greatly dependent on ATP-PC system (i.e sprinters).

Response: Yes that is the point. Nothing further needs to be added here. 

  1. Line 70-74. Please consider to modify the way you mention primary and secondary aims. In line 71 you aim to examine the differences in… It is also not clear why you only test for 1RM in only one of the groups and not both (as you did for example with VO2max). Regarding hypothesis, how do you justify this? Is there any reference to this?

Response: The aims have now been amended (see below).

“Therefore the purpose of this study was to examine whether differences in lung function and respiratory muscle strength exist between trainers that predominately engage in endurance or strength-related exercise. A secondary aim was to investigate if lung function and respiratory muscle strength were associated with VO2max, one-repetition maximum (1RM), and fat-free mass.”

1RM was only performed in the strength-trained since the squat and deadlift are highly technical lifts and it was assumed that experience for these lifts would be lacking for the endurance athletes. To reduce risk of injury and potential learning effects influencing the results it was decided that the endurance-trained participants would not perform any 1RM testing for these exercises since only 12/22 did not perform any resistance training and the remaining participants’ performing ≤2 resistance training sessions per week. Other exercises were not selected as substitutes because the squat and deadlift are the main two exercises that lead to elevated IAP and likely have a stimulatory effect on the respiratory muscles.

References have been added to all hypotheses (see below). (Note: there is no reference that can be found for VO2max)

“It was hypothesised that superior performance in the endurance-trained compared to strength-trained would be observed for indices of lung function, which is supported by evidence of enhanced lung function in athletes with endurance training experience [10,11]. Greater  respiratory muscle strength was expected in the strength-trained compared to endurance-trained based on the findings from Brown et al. [17]. Further, it was expected that lung function and respiratory muscle strength would be associated with VO2max, 1RM [19], and fat-free mass [20].”

The following has been added to the Methods explaining why only the strength-trained performed the 1RM testing:

“It was decided that the endurance-trained would not be involved in 1RM testing since 12/22 participants did not engage in regular resistance training. Also, it was likely that having novice resistance trainers perform the squat and deadlift 1RM may expose the endurance-trained participants to unnecessary risks and could confound the results due to the highly technical nature of these lifts.”

Methods Section

  1. Line 81. The study design: cross-sectional, descriptive and correlational.

Response: Changed.

  1. Line 82. Participants: level or participation in any sports specifically?

Response: Strength-trained – no; endurance-trained – has been provided.

  1. Line 87. What about the other group? Type of sports or any other detail would be important to report.

Response: The following has been added

“There were 18 participants that were considered recreational resistance trainers, five participants that had a background in powerlifting/Olympic weightlifting and one participant with a history of competing in bodybuilding contests.”

  1. Line 103-109. I don´t understand the reasoning here: the strength group was tested in: vo2max, lung function, respiratory muscles and 1RM, plus body composition. Whereas, the endurance group was tested in body composition, lung function, respiratory muscle strength and vo2max. So tests in both groups was not done in the same exact order plus one group was not tested for 1RM strength. Can the authors explain why? This is a major concern.

Reponses: Yes that is correct but how are the results of the body composition, lung function and respiratory muscle strength going to be affected since these are resting measures and with all potential confounders being controlled? The VO2max test will not be affected by prior lung function and respiratory muscle strength testing. If exercise was completed in the same session prior to any resting measure then this is a major concern. The strength testing has already been justified.

  1. Line 114. What about the other group?

Reponses: You obviously missed this sentence: “The endurance-trained group completed all testing in one visit in the order of body composition……”

  1. Line 119. Is there a reference to justify the use of this particular protocol?

Response: A reference supporting this protocol has been added.

  1. Line 157. Regarding tests, ICC and CV must be calculated to present the absolute and relative reliability of the variables obtained. In all cases, except 1RM this must be calculated and reported either in the results section or methods & design.

Response: These have now been reported in the Methods.

  1. Lines 172-173. This doesn’t make sense. It is not the same to test Squat, then Bench Press and finally Deadlift. Basically, because if you do a Bench Press initially, then participants will have to do two lower limbs exercises consecutively which in turn could affect the outcome of the test.

Response: So even if deadlift performance was affected how would squat performance be affected by the order? I will add as a limitation of the study ONLY for the deadlift performance (see below).

“It should be noted that the relationships between respiratory muscle strength and deadlift performance could have been impacted by some participants performing the squat 1RM immediately prior the deadlift 1RM. In this case, the deadlift 1RM may have been negatively affected and influenced the associations with MIP and MEP.”

  1. Lines 183-184. The use of lifting straps could also affect results. Please refer to: Jukic, I., García-Ramos, A., Malecek, J., Omcirk, D., & Tufano, J. J. (2020). The Use of Lifting Straps Alters the Entire LoadVelocity Profile During the Deadlift Exercise. The Journal of Strength & Conditioning Research, 34(12), 3331-3337.

Response: I have added as a limitation of the study (see below).

“Also, some participants used wrist straps which again could have influenced the relationships with the respiratory muscle strength measures. This is based on evidence of increased deadlift 1RM through wearing of wrist straps [35].”

  1. 189-190. Why not a T Student test? You are performing a 2 groups comparison, with no pre-post...

Response: There were numerous variables which required running direct comparison statistical tests. If conducting multiple t-tests there is an increased risk of Type I errors. A one-way analysis of variance (ANOVA) controls numerous errors so that the Type I error remains at 5% and provides greater confidence in any statistically significant result.

  1. Additionally, effect sizes with confidence intervals should be reported in conjunction with NHST. Refer to: Harrison, A. J., McErlain-Naylor, S. A., Bradshaw, E. J., Dai, B., Nunome, H., Hughes, G. T., ... & Fong, D. T. (2020). Recommendations for statistical analysis involving null hypothesis significance testing.

Response: This has been completed.

  1. Line 195. The correlations coefficients must be reported combined with their respective confidence intervals.

Response: Done (see text below and adding in Figures for respiratory muscle strength measures).

When analyzing the data of both groups a significant moderate positive relationship was found between fat-free mass and MIP for the strength-trained (r = 0.42, p = 0.04; 95% CI: 0.02 to 0.70), but not for the endurance-trained (r = 0.35, p = 0.10; 95% CI: -0.08 to 0.67). There was no significant relationship found between fat-free mass and MEP for the strength-trained (r = 0.28, p = 0.18; 95% CI: -0.14 to 0.61) or endurance-trained (r =-0.13, p = 0.56; 95% CI: -0.31 to 0.52). For MVV there was a significant strong relationship found with VO2 max (L·min-1) for the strength-trained (r = 0.55, p = 0.009; 95% CI: 0.16 to 0.79) and endurance-trained (r = 0.55, p = 0.02; 95% CI: 0.11 to 0.81). Relative VO2 max (ml·kg-1·min-1) was strongly related with MVV for the strength-trained (r = 0.63, p = 0.002; 95% CI: 0.27 to 0.83) but no relationship was found for the endurance-trained (r = 0.07, p = 0.80; 95% CI: -0.41 to 0.52). There were no significant relationships found between any other lung function indices and VO2 max.”

Results Section

  1. Line 211. Table 1. This shows a between comparison also. Table heading should also reflect this.

Response: Completed.

  1. Also, VO2 max (relative) is too low. Is there any reasoning behind this important difference?

Response: The values are representative of the characteristics of study cohort. The participants are not elite athletes and  I even provided evidence in the manuscript that: physically active but not high trained males can be shown to achieve VO2 max values of 50.1 ± 3.1 ml·kg-1·min-1 [21].” The strength-trained group were not aerobically trained and hence achieved an average relative VO2max compared to the age-matched healthy population and also for non-cyclists the VO2max/peak values can be 10-15% lower compared to a treadmill test.

  1. Figure 2. Should reflect in dashed lines the confidence intervals.

Response: The ES and 95% CI have been reported in the text for these measures and it is not deemed necessary to include in the figure.

  1. Lines 256-257. If you perform combined group correlations when participant’s characteristics is heterogeneous the outcome is spurious correlations. This is confirmed by the differences reported in the between groups comparisons. So this must be performed separately.

Response: This has now been performed and the results have been amended throughout the manuscript.

“For each individual group (i.e. analyses conducted separately for endurance- and strength-trained) the Pearson correlation coefficient was used to assess the relationships between….”

Discussion Section

  1. Line 250. As previously mentioned: the aim is not clear. This should be to examine the differences in the lung function and respiratory...

Response: This has been amended.

  1. Lines 268-269. An r = 0.47 equals an R2 = 0.22, which means this correlation explains 22% of shared variance, which too low. Also this moderate correlation is probably affected by the heterogeneity of the combination of two very different groups.

Response: Well it is still a significant finding and this remained when the correlations were run separately for each group.

  1. Lines 273-274. This is not supported by the study findings. One group showed very different characteristics compared to the other. Please see lines 261-265 where you specifically refer to this differences.

Response: Now has been changed (see below).

“It appears that differences in respiratory muscle strength in resistance trainers may be influenced by lower body strength. However, it is unclear what factors may influence the greater respiratory muscle endurance observed in the participants that predominately engage in endurance training.”

  1. Line 282. This is why author should provide ES + CI to provide additional information about practical not only statistically significant differences. This would help to understand in a more comprehensive way the present results.

Response: This has been added in the Tables and the text here has been amended.

  1. Line 291. Please check citation.

Response: This is correct.

  1. Line 304. How do authors know that endurance trained participants do not actually engage in any type of resistance training, which is actually recommended nowadays to increase performance in this particular athletic populations.

Response: 12/22 participants did not engage in any resistance training, 1/22 participants performed bodyweight resistance exercise once per week, 9/22 participants performed 1-2 days of resistance training. This has been added to Methods.

  1. Line 314. Please check and correct this sentence.

Response: Amended as shown below.

“The significant relationships between the respiratory muscle measures and 1RM for the squat and bench press may suggest that resistance exercise requiring greater lumbar stability could provide respiratory muscle strength training stimulus.”

  1. Line 316. This also explains between 4 to 16% of shared variance, which in turn explains little to nothing with respect to both variables.

Response: The following has been amended to emphasise the weaker relationships.

“Previously, significant although weaker relationships compared to the present study were found between respiratory muscle strength and knee flexor and extensor muscle strength (r = 0.21-0.41) in a mixed athlete (e.g. judo, gymnasts) cohort.”

  1. Lines 335-336. Same as previous with regards to this analysis in the combined group. Also the confidence intervals should be presented in all correlations analysis.

Response: This has been amended (see below).

“….with relative VO2 max in the strength-trained (r = 0.63, 95% CI: 0.27 to 0.83) but not the endurance-trained…”

  1. Line 340. How do authors support this? I don’t think this is part of the study aim, meaning you are measuring sensitivity of a measurement.

Response: This has been deleted and replaced with the following.

“Since exercise trainers with higher VO2 max have the ability to sustain high ventilations, it appears to be a product of exercise training type leading to specific respiratory system adaptations. Although the contribution of central and peripheral adaptations rather than respiratory system adaptations most likely influences the VO2 max (ml·kg-1·min-1) of endurance trainers [43].”

  1. Lines 341-346. Last paragraph must clearly state the study limitations. In this sense: The case of not permitting the use of weight belts could also affect 1RM estimation, so this must be pointed as a limitation (briefly).

Response: The following has been added.

“It should be noted that the relationships between respiratory muscle strength and deadlift performance could have been impacted by some participants performing the squat 1RM immediately prior the deadlift 1RM. In this case, the deadlift 1RM may have been negatively affected and influenced the associations with MIP and MEP. Also, some participants used wrist straps which again could have influenced the relationships with the respiratory muscle strength measures. This is based on evidence of increased deadlift 1RM through wearing of wrist straps [37].”

Not wearing weight belts is not a limitation of the study because this is an accessory that not everyone uses. This is needed to standardize the testing of 1RM.

  1. Conclusion Lines 359-360. These conclusions should be revised according to my earlier comments on correlations.

Response: Changed (see below)

“In male trainers that predominately engage in endurance compared to strength-related exercise there appears to be unique differences in respiratory system characteristics.. The endurance-trained group exhibited superior respiratory endurance as well as a trend towards greater lung function performance for various indices compared to the strength-trained group. In contrast, the strength-trained group displayed greater performance in the respiratory muscle strength tests. Respiratory muscle endurance was strongly related to relative VO2 max for the strength-trained but not endurance-trained. Differences in respiratory muscle strength in resistance trainers appears to be influenced by lower body strength. However, it is less clear what factors may influence the greater respiratory muscle endurance observed in endurance-trainers.”

  1. Lines 360-362. Also I still don’t understand how authors arrive to this conclusion.

Response: Again changed (see below).

“In male trainers that predominately engage in endurance compared to strength-related exercise there appears to be unique differences in respiratory system characteristics.. The endurance-trained group exhibited superior respiratory endurance as well as a trend towards greater lung function performance for various indices compared to the strength-trained group. In contrast, the strength-trained group displayed greater performance in the respiratory muscle strength tests. Respiratory muscle endurance was strongly related to relative VO2 max for the strength-trained but not endurance-trained. Differences in respiratory muscle strength in resistance trainers appears to be influenced by lower body strength. However, it is less clear what factors may influence the greater respiratory muscle endurance observed in endurance-trainers.”

I trust that the issues above have been addressed and clarified sufficiently.

Sincerely,

Dr Daniel Hackett

Reviewer 2 Report

Very well-designed research. Modern measuring equipment. An interesting and important research task for the practice of sports training. The following issues require justification in terms of methodology: - why the subjects aged 18-45 year were placed in one group - is there literature that allows one analysis to cover the respondents (with an average sports level) with a large difference in age and training experience - why in the ergometric test protocol a constant increase in load was assumed, regardless of body weight. There were significantly different body weights in the group. The work should be supplemented with practical recommendations for the construction of training programs.

Author Response

Thank you for reviewing this manuscript and providing helpful feedback.

Please find below a point-by-point response to all the comments raised. 

  1. The following issues require justification in terms of methodology:- why the subjects aged 18-45 year were placed in one group - is there literature that allows one analysis to cover the respondents (with an average sports level) with a large difference in age and training experience.

Response: The following has been added (see below). Also an ANOVA was run between groups with no differences found based on age (p>0.05).

“While pulmonary function and aerobic capacity declines between the ages of 25-80 years, an age of 45 years was considered the upper limit where the effects of age would have minimal influence on the respiratory and exercise performance of an active cohort [21].”

  1. Why in the ergometric test protocol a constant increase in load was assumed, regardless of body weight.

Response: The following reference has been added to the manuscript to support the use of this protocol.

Black, M.I.; Jones, A.M.; Kelly, J.A.; Bailey, S.J.; Vanhatalo, A. The constant work rate critical power protocol overestimates ramp incremental exercise performance. Eur J Appl Physiol 2016, 116, 2415-2422.

  1. There were significantly different body weights in the group.

Reponses: Body weight is only an issue for lung function in obese populations due to the fat located in the visceral cavity.

Sutherland, T.J.T.; McLachlan, C.R.; Sears, M.R.; Poulton, R.; Hancox, R.J. The relationship between body fat and respiratory function in young adults. European Respiratory Journal 2016, 48, 734.

Numerous correlations were run with the body composition variables and only fat-free mass was shown to influence MIP in the strength-trained but no other lung function variable or MEP.

  1. The work should be supplemented with practical recommendations for the construction of training programs.

Response: I respectfully disagree with this suggestion. This is an exploratory study therefore adding practical recommendations would be premature.

I trust that the issues above have been addressed and clarified sufficiently.

Sincerely,

Dr Daniel Hackett

Round 2

Reviewer 1 Report

Please refer to the attached document. 

Author Response

Dear Editor in Chief,

Please find below a point-by-point response to all the comments raised. 

Reviewer #1

Abstract Section

  1. Lines 14-15. Please clarify the secondary study aims. Particularly because you are analysing the correlations in 1RM in only group no in both. (Same in Lines 77-78).

Response: Thank you for this advice. These sections have been revised according to your guidance.

  1. Statistical Analysis Please provide how ICC and CV have been calculated.

Response: Done, please see below.

“The ICC was calculated using a two-way mixed model, absolute agreement, single rater/measurement [26]. Coefficient of variation (CV) was calculated by dividing standard deviation (SD) by the mean of trials × 100.”

  1. 189-190. Please consider performing an independent T Student test. Student's t-test is used when two independent groups are compared, while the ANOVA extends the t-test to more than two groups. You are performing a 2 groups comparison, independently of how many variables you need to compare. If there was a pre-post, then an ANOVA would be the correct one... Additionally, previous to an ANOVA you must check for homogeneity (which you have not done) through Levenne’s test.

Response: Yes an independent Student’s T-test has now been used and the results have been amended. However, there was very minimal change from my previous analyses.

  1. Line 195. The correlations coefficients must be reported combined with their respective confidence intervals.

Response: Now amended.

  1. Results Section Figure 2. Please consider to include in dashed lines the confidence intervals, which will provide a clear (visually) interpretation on how the data points are scattered across the regression line.

Response: This has now been completed.

  1. Discussion Section Line 321-322. The reported values are not strong ES, please refer to the criteria provided in the statistics section. Revise throughout.

Response: Only one instance of this and it has now been amended.

Thank you again for your recommendations and suggestions which has improved this paper.

Sincerely,

Dr Daniel Hackett